# Systemic cisplatin increases the number of patients showing positive off-frequency masking audiometry

**Harukazu Hiraumi** \*, **Shin-ichi Oikawa, Kiyoto Shiga, Hiroaki Sato**

Department of Otolaryngology—Head and Neck Surgery, Iwate Medical University, Yahaba, Shiwa, Iwate, Japan

\* hhiraumi@ent.kuhp.kyoto-u.ac.jp

## Abstract

### Objective

The study aimed to evaluate the effect of systemic cisplatin administration on off-frequency masking audiometry.

### Methods

Among 26 patients receiving systemic cisplatin, 48 ears were included in the analysis. All patients underwent pure-tone audiometry with ipsilateral narrow-band masking noise (off-frequency masking audiometry). In the off-frequency masking audiometry, 70 dBHL band-pass noise (center frequency 1000 Hz, 1/3 octave bandwidth) was administered to the tested ear. The acquired thresholds were compared to those of standard pure-tone audiometry, and threshold elevations greater than 10 dB were regarded as significant. The number of patients showing abnormal threshold elevation was compared between before and after the cisplatin administration.

### Results

Before cisplatin administration, 91.7, 93.8, 97.9, and 93.8% of ears showed normal off-frequency masking audiometry outcomes at 125, 250, 6000, and 8000 Hz, respectively. After cisplatin administration, a higher number of patients showed abnormal off-frequency masking audiometry outcomes. This change was more prominent with increasing doses of cisplatin. After the cisplatin administration of $100 \sim 200$ mg/m$^2$, the prevalence of patients with normal off-frequency masking audiometry outcomes was 77.3, 70.5, 90.9, and 88.6% at 125, 250, 6000, and 8000 Hz, respectively. At 250 Hz, the change was statistically significant (p = 0.01, chi-squared test).

**Data Availability Statement:** All relevant data are within the Supporting Information files.

**Funding:** This work was supported by JSPS KAKENHI (grant number JP22K09748). The funders had no role in study design, data collection

and analysis, decision to publish, or preparation of
the manuscript.

**Competing interests:** The authors have declared
that no competing interests exist.

**Abbreviations:** CDDP, cisplatin; DPOAE, distortion
product otoacoustic emission; TEN, threshold-
equalizing noise.

## 1. Introduction

Hearing disorders are among the most prevalent disabilities. In addition to its negative effect on communication, hearing disorders are a serious risk factor for dementia [1]. Hearing disorders are usually evaluated using auditory tests, including pure-tone audiometry, auditory brainstem response, and otoacoustic emission. Most of these tests are designed to measure auditory thresholds. Since auditory thresholds are highly influenced by the function of the outer hair cells [2], these tests mainly evaluate auditory efferent function, and auditory afferent disorder has long been neglected except for profound sensorineural hearing loss requiring cochlear implantation.

Recently, auditory afferent disorders have become the focus of research. Hidden hearing loss is a pathological condition reported by Kujawa and Liberman [3]. They conducted cochlear functional assays and confocal imaging of the inner ear of mice with transient noise-induced hearing loss. They discovered that the amplitude of the evoked response by intense high-frequency sounds permanently decreased despite the preserved pure-tone thresholds. Histologically, they observed acute loss of afferent nerve terminals and delayed degeneration of the cochlear nerve [3]. Since these reports, several pathologies in the auditory afferent systems have been reported in animal models. All animal models showed normal or near-normal hearing thresholds and deteriorated responses to intense sounds. For instance, degeneration of the ribbon synapse similar to that of hidden hearing loss was observed in an aged mouse model [4]. Since elderly human histology shows similar findings [5], the degeneration of the ribbon synapse is regarded as a candidate for the underlying mechanism of hearing disability despite normal or near-normal hearing thresholds in the elderly. Wan and Corfas used a mouse model to report that transient auditory nerve demyelination causes the same symptoms as those of hidden hearing loss. Histologically, this mouse model showed a disrupted hemi-node structure of the cochlear nerve [6]. Lobarinas et al. conducted hearing tests in a chinchilla model with selected inner hair cell loss. The study revealed that pure-tone thresholds were preserved, whereas the thresholds increased significantly when they were tested in the presence of competing noise [7]. Interestingly, this elevation of the threshold was observed over a wide range of frequencies when narrow-band noise was used as a masker (off-frequency masking).

This inner hair cell-loss model was developed using platinum-based anticancer drugs that damage different inner ear organs in different species. For example, carboplatin damages the outer hair cells in guinea pigs [8], whereas it damages the inner hair cells in chinchillas [9]. In humans, only a limited number of histopathological studies have reported the ototoxicity of platinum-based anti-cancer drugs. These reports agree that cisplatin (CDDP) first damages the outer hair cells in the basal turn [10]. Based on this, the ototoxicity of CDDP has been evaluated using pure-tone audiometry and distortion product otoacoustic emission (DPOAE), which reflect outer hair cell damage [11]. Other inner ear organs in the auditory afferent systems are also reported to be damaged by CDDP. For instance, CDDP has been reported to damage spiral ganglion cells predominantly in the upper turns [12]. It also damages the inner hair cells inconsistently, although the damage is milder than that of the outer hair cells [13]. These reports suggest that the effect of off-frequency masking can be observed in patients receiving CDDP, especially at low frequencies coded in the upper turns of the cochlea.

To test this hypothesis, we conducted off-frequency masking audiometry in patients with head and neck cancer before and after CDDP treatment.

## 2. Methods

### 2.1 Participants

Between November 2020 and January 2021, 26 patients with head and neck cancer were administered chemotherapy, including CDDP (age, 50–79 years old, average 66.9 years old, 23 males and 3 females). In this study, we evaluated the bilateral ears independently. Since middle and external ear pathologies occurred during treatment in 4 ears (otitis media with effusion in 2 ears, occlusion on the external auditory canal due to parotid cancer in 1 ear, and transient low-tone conductive hearing loss with unknown etiology in 1 ear), these were excluded; finally, 48 ears were included in the analysis. All study participants provided verbal informed consent, since all the tests were within the coverage of public insurance of Japan. It was explained to the participants that off-frequency masking audiometry is approved by public insurance but its clinical significance in patients administered CDDP is unknown. The study design was approved by the Ethics Committee of Iwate Medical University (MH2022-125), in accordance with the Declaration of Helsinki and the Ethical Guidelines for Medical and Health Research Involving Human Subjects issued by the Ministry of Health, Labour and Welfare of Japan.

### 2.2 Audiological tests

All patients underwent standard pure-tone audiometry and DPOAE. Off-frequency masking audiometry is a type of noise audiometry. In this test, a pure-tone threshold is measured under a masking sound administered to the tested ear. The center frequency of the masking sound is set outside the tuning curve of the signal sound. Noise audiometry including off-frequency masking audiometry is intended to detect a recruitment phenomenon. However, the practical significance of these tests is low [14], and off-frequency masking audiometry is usually used only for a non-clinical purpose; for example, to measure the psychophysiological tuning curve [15], to measure the temporal integration patten under a masking sound [16], or to measure the masking level difference [17]. In this study, a 70-dBHL narrow band noise (center frequency 1000 Hz, 1/3 octave bandwidth) was administered to the tested ear. Pure-tone thresholds with competing narrow-band noise were measured between 125 and 8000 Hz. The acquired thresholds were compared with those obtained using standard pure-tone audiometry. Threshold elevations greater than 10 dB were considered significant (off-frequency masking positive), according to the criteria for different types of noise audiometry [14, 18]. These tests were conducted before initiating chemotherapy and repeated 1 week after each cycle of CDDP administration. The average time period between initial and final testing was 40 days (35–47 days). The prevalence of off-frequency masking (+) patients was analyzed according to the cumulative dose of CDDP. In this study, only eight patients received CDDP of more than 200 mg/m$^2$, and the cumulative CDDP dose was categorized into two classes (<100 mg/m$^2$ and 101–200 mg/m$^2$).

Conventional auditory tests (pure-tone audiometry and DPOAE) were also analyzed along with the cumulative dose of CDDP. Hearing deterioration as per pure-tone audiometry is defined as a threshold shift greater than 10 dB. The DPOAE was defined as positive when the amplitude was 6 dB above the noise floor. The prevalence of hearing deterioration in the pure-tone audiometry and the prevalence of negative DPOAE were compared between the off-frequency masking (+) patients and off-frequency masking (-) patients.

### 2.3 Statistical analyses

All statistical analyses were performed using SPSS software (IBM SPSS Statistics 24 for Windows, Advanced Analytics Inc., Tokyo, Japan). The pure-tone thresholds were tested with

analyses of variance (ANOVA). The prevalence of ears with positive DPOAE and the prevalence of off-frequency masking (+) ears was tested with Fisher's exact test if the data matrix is $2 \times 2$, In the other cases, the chi-squared test was used to compare groups. A p-value of 0.05 was considered statistically significant.

## 3. Results

Pure-tone audiometry revealed high-frequency hearing loss before the initiation of chemotherapy. The hearing threshold at high frequencies was significantly elevated with the cumulative dose of cisplatin ($p = 0.044$ for 6000 Hz, $p = 0.003$ for 8000 Hz, ANOVA); however, the hearing level at low frequencies did not change ($p = 0.956$, 125 Hz; $p = 0.997$, 250 Hz; and $p = 0.860$, 500 Hz, ANOVA). The average pure-tone audiogram is shown in Fig 1. The DPOAEs were positive at any frequency in 35 ears before the administration of CDDP. The prevalence of positive DPOAE tended to be lower in high frequencies after the administration of CDDP; however, the change was not significant ($p = 0.348$ for 1000 Hz, $p = 0.823$ for 1400 Hz, $p = 0.886$ for 2000 Hz, $p = 0.218$ for 2800 Hz, $p = 0.076$ for 4000 Hz, and $p = 0.192$ for 6000 Hz, chi-squared test).

Before the administration of CDDP, thresholds for 125, 250, 6000, and 8000 Hz were not affected by narrow-band noise in >90% of the tested ears (91.7, 93.8, 97.9, and 93.8% at 125, 250, 6000, and 8000 Hz, respectively) (Fig 2A), and these frequencies were regarded as off-frequencies for the narrow-band masking noise. After the administration of CDDP, the prevalence of off-frequency masking (+) ears was higher than that before CDDP administration at low frequencies (Fig 2B and 2C). Finally, >20% of the ears were positive for off-frequency masking at 125 and 250 Hz (22.7% and 29.5%, respectively). At 250 Hz, this change was statistically significant ($p = 0.014$, Cramer's V = 0.248, chi-squared test). Post-hoc analysis using Fisher's exact test with Bonferroni adjustment showed that the prevalence of off-frequency masking (+) ears receiving 101–200 mg/m$^2$ of CDDP was significantly higher than that before CDDP administration ($p = 0.015$ after Bonferroni adjustment, Cramer's V = 0.307, Fisher's exact test).

Similar trends were observed at high frequencies but were not as prominent as those at low frequencies (9.1% at 6000 Hz and 11.4% at 8000 Hz).

Among the ears with off-frequency masking (+) at 250 Hz, none showed threshold elevation at 125–500 Hz on pure-tone audiometry.

The prevalence of positive DPOAE after CDDP administration was higher in off-frequency masking (+) ears than in those with negative off-frequency masking at 1000 Hz and 2000 Hz ($p = 0.037$ for 1000 Hz, and $p = 0.003$ for 2000 Hz, Fisher's exact test). The prevalence of pure-tone threshold elevation after administration of cisplatin was not different between ears with and without the effect of off-frequency masking ($p = 0.544$ for 125 Hz, $p = 0.301$ for 500 Hz, $p = 1.000$ for 1000 Hz, $p = 0.652$ for 2000 Hz, $p = 0.404$ for 2000 Hz, $p = 0.461$ for 3000 Hz, $p = 1.000$ for 4000 Hz, $p = 0.737$ for 6000 Hz, and $p = 1.000$ for 8000 Hz, Fisher's exact test).

## 4. Discussion

Our analyses revealed that the masking effect of band-pass noise was small at low frequencies and large at high frequencies before CDDP administration, which is consistent with previous reports [15]. After the administration of CDDP, significantly more ears showed threshold elevation at low frequencies with off-frequency masking. In ears showing a positive off-frequency masking effect, the pure-tone audiogram and DPOAE were less or similarly damaged than those of the off-frequency masking (-) ears. These results suggest that positive off-frequency masking is derived from a different mechanism from outer hair cell damage.

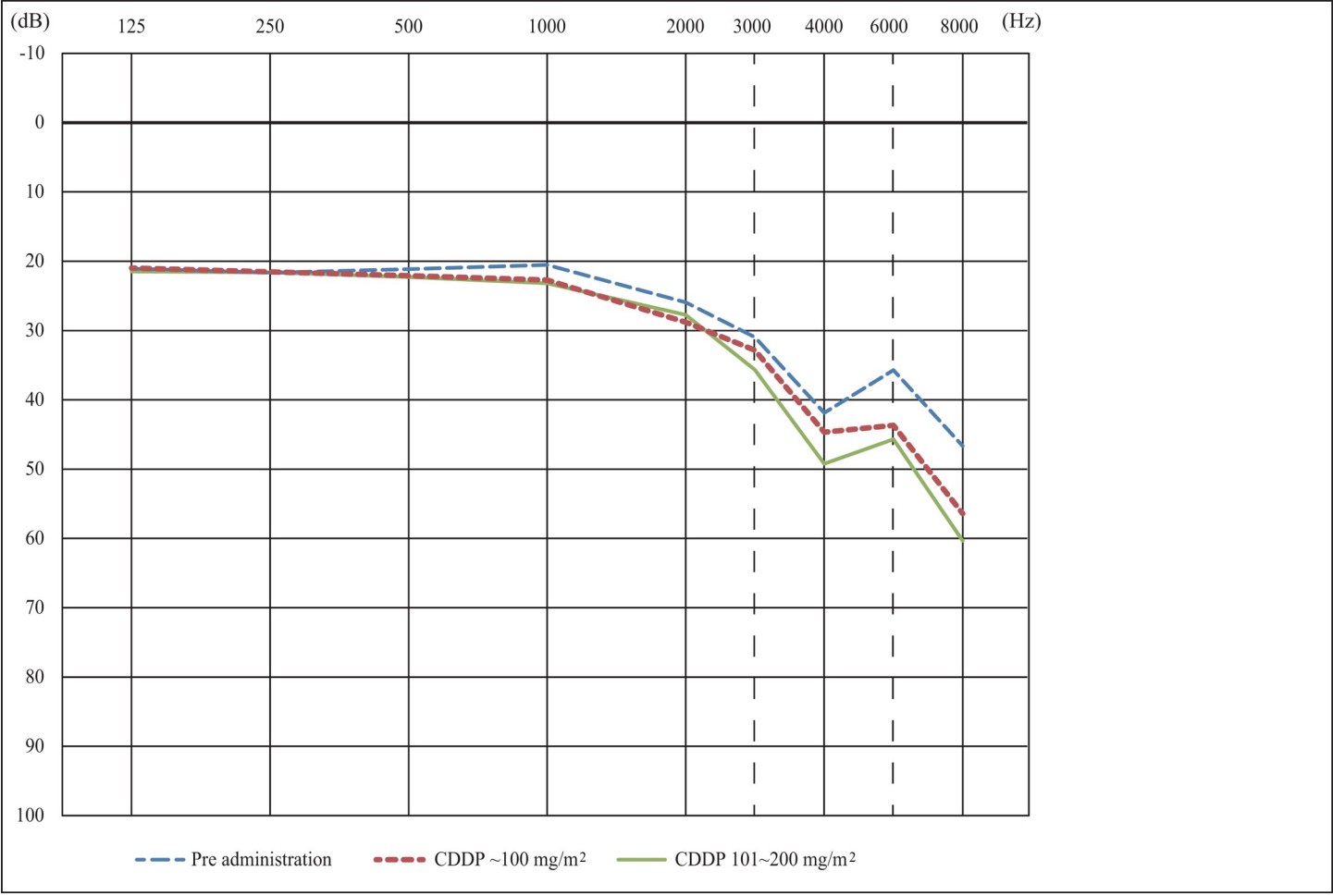

**Fig 1. The pure-tone average before and after the administration of cisplatin.** The averaged audiogram before the administration of cisplatin showed a down-sloping type sensorineural hearing loss. The pure-tone thresholds deteriorated significantly with the cumulative dose of cisplatin in the 6000–8000 Hz range.

In humans, pure-tone audiometry with ipsilateral-masking noise (noise audiometry) has long been regarded as a test with limited clinical significance [14]. Recently, some studies have reported that patients with auditory afferent disorder show abnormal findings on noise audiometry. In normal-hearing subjects and patients with ordinary sensorineural hearing loss, the pure-tone thresholds are not changed by masking noise lower than the detection thresholds; however, in patients with auditory neuropathy spectrum disorder, wide-band noise elevates the pure-tone thresholds [19]. In free-field audiometry, wide-band noise elevated the pure-tone thresholds at low and high frequencies in the normal-hearing elderly, which was not observed in young participants [20]. The threshold-equalizing noise (TEN) test is a kind of noise audiometry that was developed to detect a limited and severe loss of inner hair cells (dead region) [21]. The pure-tone audiogram cannot detect the dead region since the inner hair cells surrounding the dead region respond to the stimuli. Damage to outer hair cells promotes stimulation of the surrounding inner hair cells through reduced tonotopicity [22]. In the TEN test, pure-tone audiometry is conducted under wideband noise (TEN). A dead region is indicated when the TEN elevates the pure-tone threshold at a particular frequency. The TEN

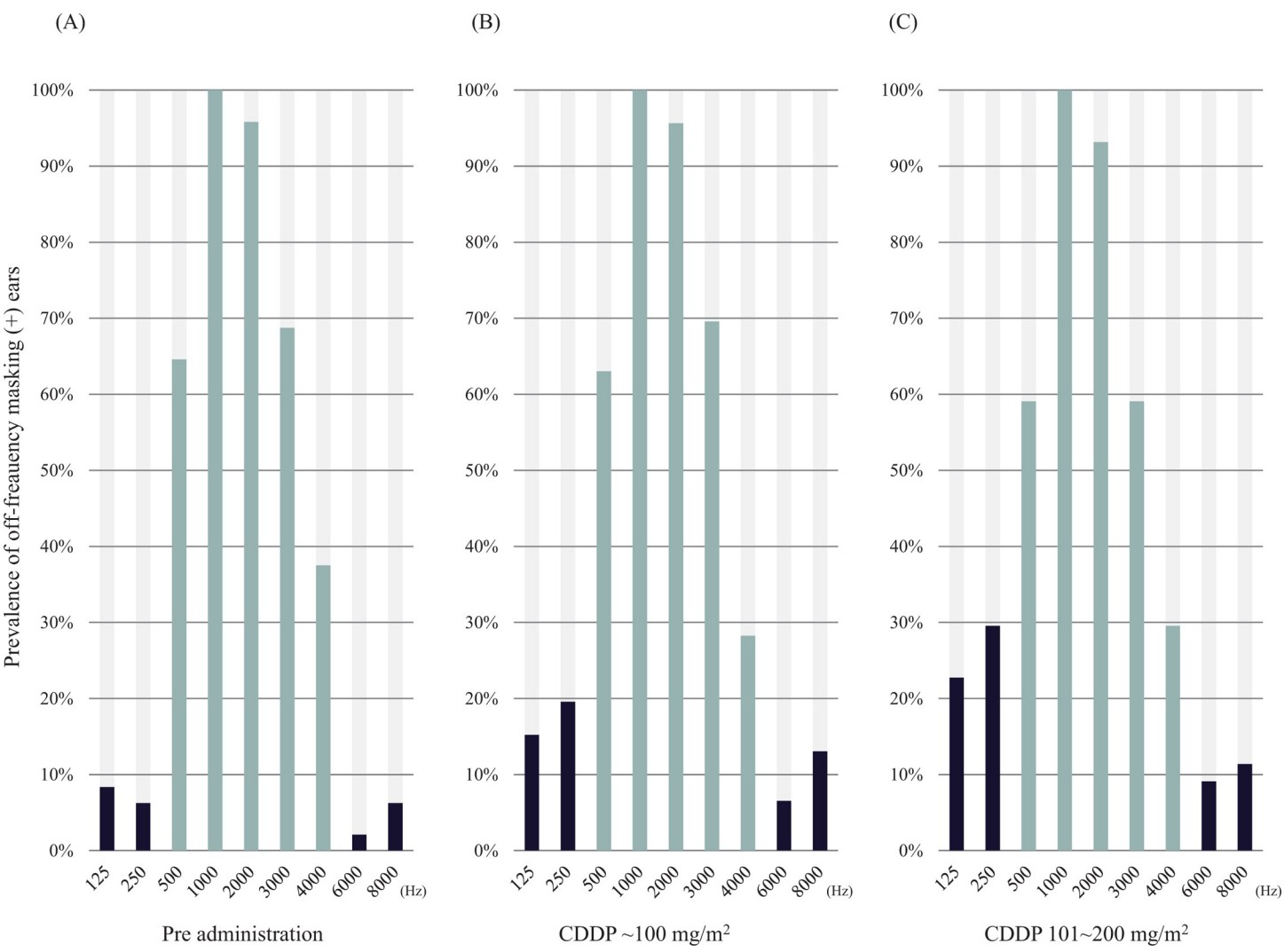

(A) Pre administration

(B) CDDP ~100 mg/m²

(C) CDDP 101~200 mg/m²

**Fig 2. The prevalence of ears showing an abnormal threshold shift with a narrow band noise.** The prevalence of ears showing threshold shift with narrow-band noise (center frequency of 1000 Hz) is shown. Between 500 and 4000 Hz, the narrow-band noise elevated the pure-tone threshold (on-frequency masking, colored pale blue). At 125, 250, 6000, and 8000 Hz, only <10% of the patients showed a threshold shift with narrow band noise before the administration of cisplatin (off-frequency masking, colored black) (A). After the administration of cisplatin, threshold elevation at 125 and 250 Hz with narrow-band noise was observed in more ears (B). This tendency became more prominent at higher doses of cisplatin, and the change at 250 Hz was statistically significant (C).

test has been proposed to be a diagnostic tool for other auditory afferent disorders, including hidden hearing loss, although its significance needs to be further explored [23].

Recently, Schultz et al. reported that patients showed abnormal threshold shifts in the TEN test after the administration of CDDP [24]. These patients showed only mild-to-moderate hearing loss; however, abnormal TEN test results were observed over a wide range of frequencies. This result is very interesting yet confusing since systemic CDDP administration is not likely to affect the limited area of inner hair cells. Instead, CDDP could induce a wide range of auditory afferent disorders, including the inner hair cells and auditory afferent disorder resulting in the threshold elevation with the coexisting masking noise, similar to the present study.

In the present study, we observed an abnormal threshold elevation with off-frequency masking that could not be explained by outer hair cell damage. However, the underlying mechanism of this result has not been determined. Two mechanisms are speculated to explain

the results of our study. One is the loss of inhibition of the auditory pathway which may widen the masking area of narrow-band noise, resulting in positive off-frequency masking [7]. In humans, acute inner ear injury enhances the activity of the auditory cortex, which is regarded as a result of the loss of inhibition in the brainstem and/or auditory cortex [25]. This loss of inhibition at some level of the auditory afferent system could be the underlying mechanism of off-frequency masking in the present study. The other possible hypothesis is that the pure-tone with off-frequency masking elicites distinct auditory fibers from those involved in the pure-tone audiometry. The auditory nerve fibers can be divided into three groups according to the spontaneous discharge rate and each group has different properties. For example, low spontaneous-discharge-rate fibers have a higher threshold and a higher saturation point [26], and selective damage to low spontaneous-discharge-rate fibers is reported in hidden hearing loss [27]. It is possible that CDDP selectively damages the nerve fibers responsible to decide the thresholds in off-frequency masking.

There are some limitations in this study. For example, we measured DPOAE to evaluate the function of the outer hair cells; however, DPOAE does not reflect the outer hair cell function at low frequencies. Another limitation is that we analyzed two ears from each participant because some participants showed unilateral conductive hearing loss. Since CDDP is expected to affect two ears similarly, the analysis of both ears may exaggerate the effect of CDDP. Analysis of the participants (the number of participants showing off-frequency masking (+) in at least one ear) resulted in similar results (significant difference at 250 Hz), and we think the present result is acceptable. The third problem is that audiometry is a subjective test. We repeated audiometry during CDDP administration, and it is possible that the participants learned to perform better in audiometry. Otherwise, the administration of CDDP might have affected the cognitive skills and alertness of the participants, which may result in poor hearing thresholds. In this study, all tests were conducted by experienced and certified audiologists and no change was observed in the pure-tone audiogram at lower frequencies during CDDP administration. Therefore, we think that the threshold in the off-frequency masking is reliable. The other limitation is that inner hair cell damage was not directly proven in ears with positive off-frequency masking. We hypothesized the occurrence of inner hair cell damage after CDDP administration, but injury to other organs in the auditory afferent system can cause similar results. In human studies, it seems challenging to precisely locate the focus of the pathology, and it may be reasonable to categorize these pathologies as auditory afferent disorders. Further animal studies and histopathological studies of human temporal bones are needed to prove these problems.

## 5. Conclusion

After systemic CDDP administration, narrow band noise with a 1000-Hz center frequency elevated the threshold for 250-Hz pure-tone audiometry in significantly high numbers of patients. The pure-tone thresholds at low frequencies and DPOAE were not influenced by CDDP. This suggests that systemic CDDP affects auditory afferent pathways.

## Supporting information

**S1 Table. Audiological test results.**
(XLSX)

## Acknowledgments

We would like to thank Editage (www.editage.com) for the English language editing.

## Author Contributions

**Conceptualization:** Harukazu Hiraumi, Hiroaki Sato.

**Data curation:** Harukazu Hiraumi, Shin-ichi Oikawa, Kiyoto Shiga, Hiroaki Sato.

**Formal analysis:** Harukazu Hiraumi.

**Funding acquisition:** Harukazu Hiraumi.

**Investigation:** Harukazu Hiraumi.

**Methodology:** Harukazu Hiraumi, Shin-ichi Oikawa.

**Project administration:** Harukazu Hiraumi.

**Supervision:** Harukazu Hiraumi, Kiyoto Shiga, Hiroaki Sato.

**Validation:** Harukazu Hiraumi, Shin-ichi Oikawa, Kiyoto Shiga, Hiroaki Sato.

**Writing – original draft:** Harukazu Hiraumi.

**Writing – review & editing:** Harukazu Hiraumi, Shin-ichi Oikawa, Kiyoto Shiga, Hiroaki Sato.

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
