## [Decision Letter · Decision Letter 0]

6 Mar 2023

PONE-D-23-02457Systemic Cisplatin Increases the Number of Patients Showing Positive Off-frequency Masking AudiometryPLOS ONE

Dear Dr. Hiraumi,

Thank you for submitting your manuscript to PLOS ONE. After careful consideration, we feel that it has merit but does not fully meet PLOS ONE’s publication criteria as it currently stands. Therefore, we invite you to submit a revised version of the manuscript that addresses the points raised during the review process.

We look forward to receiving your revised manuscript.

Kind regards,

Antonino Maniaci

Academic Editor

PLOS ONE

Journal Requirements:

a) Did participants provide their written or verbal informed consent to participate in this study?

Additional Editor Comments:

Please edit and perform all the revisions required by the reviewer. English editing is also required. Adapt the paper according the latest guidelines form the Equator database.

Best regards

Reviewers' comments:

Reviewer's Responses to Questions

**Comments to the Author**

1. Is the manuscript technically sound, and do the data support the conclusions?

Reviewer #1: Partly

Reviewer #2: Yes

2. Has the statistical analysis been performed appropriately and rigorously? 

Reviewer #1: No

Reviewer #2: Yes

3. Have the authors made all data underlying the findings in their manuscript fully available?

Reviewer #1: No

Reviewer #2: Yes

4. Is the manuscript presented in an intelligible fashion and written in standard English?

Reviewer #1: No

Reviewer #2: Yes

5. Review Comments to the Author

Reviewer #1: The authors report on their study of the effects of cisplatin on hearing. Their cohort was 26 patients, and they utilised data from 48 ears. Rather than (actually as well as) standard audiometry, they used off-frequency masking audiometry to assess hearing.

This is an interesting topic, and the authors have clearly thought through the principles of what they are investigating. I do, however, have some concerns on the analysis and clarity of some sections.

The manuscript is need of attention from an expert English editor to correct the sentence structure in places and also the spelling here and there.

Specific comments.

Abstract:

L21: fix typo

L27: it would be good to mention the number or percentage.

Introduction:

L45: It is not clear to me what ‘mild to moderate’ auditory afferent disorder is. Please reference this.

L48: Please reference properly.

2.1 Participants: Taking both ears from some of the participants may be a problem. It should at least be stated here why the data from both ears was not included?

On this topic, the description of the methods does not say anything about how the data from the two ears were handled. I assume analysis considered participants, and not ears when assessing change in status of hearing. This needs to be carefully explained. From L130 it seems that ears were analysed; however, seeing cisplatin can be expected to affect hearing bilaterally, the inclusion of both ear biases the analysis. For example, if those with data from two ears had slightly more loss than those with data from one ear, then the difference is exaggerated. It would also influence the potential effect of does. L165 seems to address this point, but it is not clear whether this section just assesses the two ears, but the rest of the analysis is using ears.

L103: I am not familiar with the technique of off-frequency masking audiometry. Please clarify what a threshold elevation means. And please justify and/or reference why 10dB was chosen as significant.

The time period between initial testing and final testing needs to be included.

L108: I am not a chemist, but shouldn’t the dose be in cubic metres, not square metres? I assume it is the weight of the person, not a measure of area.

L112: Why was 10dB change considered significant?

I would caution against using just p-values to assess significance. Effect size is more important.

There is a learning curve associated with audiometry; how did the study team deal with this?

Figure 2: The vertical axes should be labelled. The use of black for some bars is not apparent. I wonder whether a table might be better to report these data.

Line 145 onwards: Be careful not to put too much weight on non-significant changes. If the change or difference is not significant, then there is no change/difference.

I see that Fisher’s exact test was used, probably because numbers were low. Section 2.3 needs a bit more depth and rationale for choosing these tests. When were they applied? When chi-squared and when Fisher’s?

Audiometry is a behavioural test. I wonder whether people undergoing chemotherapy may feel generally unwell (they do – I know because I have observed this at close hand over the past 3 months). This may affect their cognitive skills and alertness, etc. in order to complete audiometry. This limitation should be acknowledged. I think a 5 or 10 dB shift is not unreasonable in this situation. On the other hand, I acknowledge that no change was observed in the pure tone audiogram at lower frequencies.

L229: this sentence needs to be reworded.

L242: It is not clear how this conclusion can be made. It is good to consider the clinical implications, but there was nothing in this study that looked at better management of patients. I suggest it be removed.

Reviewer #2: The present paper is a well conducted study covering a very very specific topic.

I would only recommend a review of the English language both for typos and to clarify some lines: lines 181-182; line 213 double "the"; line 224 "fibers is"; 232 precisely

6. PLOS authors have the option to publish the peer review history of their article (what does this mean?). If published, this will include your full peer review and any attached files.

Reviewer #1: No

Reviewer #2: No

---

## [Author Response · Author response to Decision Letter 0]

9 Apr 2023

Reviewer #1: The authors report on their study of the effects of cisplatin on hearing. Their cohort was 26 patients, and they utilised data from 48 ears. Rather than (actually as well as) standard audiometry, they used off-frequency masking audiometry to assess hearing.

This is an interesting topic, and the authors have clearly thought through the principles of what they are investigating. I do, however, have some concerns on the analysis and clarity of some sections.

The manuscript is need of attention from an expert English editor to correct the sentence structure in places and also the spelling here and there.

Specific comments.

Abstract:

L21: fix typo

Answer

Thank you for pointing out this typo.

The paper has been revised by a professional proofreader.

L27: it would be good to mention the number or percentage.

Answer

Thank you for the advice. The percentages of the participants were added.

L26-28

“Before cisplatin administration, 91.7, 93.8, 97.9, and 93.8% of ears showed normal off-frequency masking audiometry outcomes at 125, 250, 6000, and 8000 Hz, respectively.”

L30-32

“…the prevalence of patients with normal off-frequency masking audiometry outcomes was 77.3, 70.5, 90.9, and 88.6% at 125, 250, 6000, and 8000 Hz, respectively.”

L163-165

“Before the administration of CDDP, thresholds for 125, 250, 6000, and 8000 Hz were not affected by narrow-band noise in >90% of the tested ears (91.7, 93.8, 97.9, and 93.8% at 125, 250, 6000, and 8000 Hz, respectively)”

Introduction:

L45: It is not clear to me what ‘mild to moderate’ auditory afferent disorder is. Please reference this.

Answer

The description was changed to make it more accurate.

L49-50

“…and auditory afferent disorder has long been neglected except for profound sensorineural hearing loss requiring cochlear implantation.” 

L48: Please reference properly.

Answer

L52

“reported by Kujawa and Liberman [3]. They…”

2.1 Participants: Taking both ears from some of the participants may be a problem. It should at least be stated here why the data from both ears was not included?

Answer

L95-98

“Since middle and external ear pathologies occurred during treatment in 4 ears (otitis media with effusion in 2 ears, occlusion on the external auditory canal due to parotid cancer in 1 ear, and transient low-tone conductive hearing loss with unknown etiology in 1 ear), these were excluded; finally, 48 ears were included in the analysis.”

On this topic, the description of the methods does not say anything about how the data from the two ears were handled. I assume analysis considered participants, and not ears when assessing change in status of hearing. This needs to be carefully explained. From L130 it seems that ears were analysed; however, seeing cisplatin can be expected to affect hearing bilaterally, the inclusion of both ear biases the analysis. For example, if those with data from two ears had slightly more loss than those with data from one ear, then the difference is exaggerated. It would also influence the potential effect of does. L165 seems to address this point, but it is not clear whether this section just assesses the two ears, but the rest of the analysis is using ears.

Answer

Thank you for the important advice. We evaluated bilateral ears independently instead of evaluating the participants. Some patients had unilateral conductive hearing loss, and they are not well analyzed as participants.

We made it clear that we analyzed two ears separately, and this may bias the results.

L94

“In this study, we evaluated the bilateral ears independently.”

L255-260

“Another limitation is that we analyzed two ears from each participant because some participants showed unilateral conductive hearing loss. Since CDDP is expected to affect two ears similarly, the analysis of both ears may exaggerate the effect of CDDP. Analysis of the participants (the number of participants showing off-frequency masking (+) in at least one ear) resulted in similar results (significant difference at 250 Hz), and we think the present result is acceptable.”

L103: I am not familiar with the technique of off-frequency masking audiometry. Please clarify what a threshold elevation means. And please justify and/or reference why 10dB was chosen as significant.

Answer

I provided a further explanation of off-frequency masking audiometry. 

In the clinical setting, a 10-dB change is regarded as significant. According to previous studies, the test-retest reliability for audiometry is <5 dB. In off-frequency masking audiometry, the test-retest reliability is unknown, but other noise audiometry tests regard 10-dB changes as significant. I have cited some references.

L109-118

“Off-frequency masking audiometry is a type of noise audiometry. In this test, a pure-tone threshold is measured under a masking sound administered to the tested ear. The center frequency of the masking sound is set outside the tuning curve of the signal sound. Noise audiometry including off-frequency masking audiometry is intended to detect a recruitment phenomenon. However, the practical significance of these tests is low [14], and off-frequency masking audiometry is usually used only for a non-clinical purpose; for example, to measure the psychophysiological tuning curve [15], to measure the temporal integration patten under a masking sound [16], or to measure the masking level difference [17].”

L122-123

“Threshold elevations greater than 10 dB were considered significant (off-frequency masking positive), according to the criteria for different types of noise audiometry [14, 18].”

The time period between initial testing and final testing needs to be included.

Answer

L125-126

“The average time period between initial and final testing was 40 days (35–47 days).”

L108: I am not a chemist, but shouldn’t the dose be in cubic metres, not square metres? I assume it is the weight of the person, not a measure of area.

Answer

In systemic chemotherapy, the dose is usually determined according to the body surface area (BSA). 

L112: Why was 10dB change considered significant?

Answer

As I described before, a 10-dB change is regarded as significant in the clinical setting.

L122-123

“Threshold elevations greater than 10 dB were considered significant (off-frequency masking positive), according to the criteria for different types of noise audiometry [14, 18].”

I would caution against using just p-values to assess significance. Effect size is more important.

Answer

I added the effect size. 

L169-170

“At 250 Hz, this change was statistically significant (p = 0.014, Cramer's V = 0.25, chi-squared test).

L171-174

“Post-hoc analysis using Fisher’s exact test with Bonferroni adjustment showed that the prevalence of off-frequency masking (+) ears receiving 101–200 mg/m2 of CDDP was significantly higher than that before CDDP administration (p = 0.015 after Bonferroni adjustment, Cramer's V = 0.31, Fisher’s exact test).”

There is a learning curve associated with audiometry; how did the study team deal with this?

Audiometry is a behavioral test. I wonder whether people undergoing chemotherapy may feel generally unwell (they do – I know because I have observed this at close hand over the past 3 months). This may affect their cognitive skills and alertness, etc. in order to complete audiometry. This limitation should be acknowledged. I think a 5 or 10 dB shift is not unreasonable in this situation. On the other hand, I acknowledge that no change was observed in the pure tone audiogram at lower frequencies.

Answer

These two questions are about the reliability of the audiometry.

All the tests were conducted by experienced and certified audiologists. We did not find improvement of the threshold, and a learning curve was not observed.

The effect of cognitive ability is still controversial. A previous study reports that the audiogram is reliable in subjects with mild dementia (McClannahan et al., 2021), but the amount of the acceptable error was not mentioned. In the TEN test, a 10-dB threshold change is regarded as significant, and this test was applied to the participants receiving chemotherapy (Schultz et al., 2019). 

We referred to the reliability of the tests.

L260-268

“The third problem is that audiometry is a subjective test. We repeated audiometry during CDDP administration, and it is possible that the participants learned to perform better in audiometry. Otherwise, the administration of CDDP might have affected the cognitive skills and alertness of the participants, which may result in poor hearing thresholds. In this study, all tests were conducted by experienced and certified audiologists and no change was observed in the pure-tone audiogram at lower frequencies during CDDP administration. Therefore, we think that the threshold in the off-frequency masking is reliable.”

Figure 2: The vertical axes should be labelled. The use of black for some bars is not apparent. I wonder whether a table might be better to report these data.

Answer

I labelled the vertical axis. In this figure, bars for 500-4000 Hz means “on-frequency” masking, and they are colored pale blue. The legends and bar colors were changed to make this clearer.

To differentiate “off-frequency” and “on -frequency”, I would like to use a figure instead of a table.

L180-188

“The prevalence of ears showing threshold shift with narrow-band noise (center frequency of 1000 Hz) is shown. Between 500 and 4000 Hz, the narrow-band noise elevated the pure-tone threshold (on-frequency masking, colored pale blue). At 125, 250, 6000, and 8000 Hz, only <10% of the patients showed a threshold shift with narrow band noise before the administration of cisplatin (off-frequency masking, colored black) (A). After the administration of cisplatin, threshold elevation at 125 and 250 Hz with narrow-band noise was observed in more ears (B). This tendency became more prominent at higher doses of cisplatin, and the change at 250 Hz was statistically significant (C).”

Line 145 onwards: Be careful not to put too much weight on non-significant changes. If the change or difference is not significant, then there is no change/difference.

Answer

Thank you for this important advice. I was not clear about the description.

L166-168

“After the administration of CDDP, the prevalence of off-frequency masking (+) ears was higher than that before CDDP administration at low frequencies (Fig 2-B, C). This change was more prominent with higher doses of CDDP administration”

I see that Fisher’s exact test was used, probably because numbers were low. Section 2.3 needs a bit more depth and rationale for choosing these tests. When were they applied? When chi-squared and when Fisher’s?

Answer

I am sorry for the poor explanation. In off-frequency masking audiometry, the matrix is 3 × 2, and the chi-squared test was used. Fisher's exact test was applied in the post-hoc analysis.

L141-143

“The prevalence of ears with positive DPOAE and the prevalence of off-frequency masking (+) ears was tested with Fisher's exact test if the data matrix is 2 × 2, In the other cases, the chi-squared test was used to compare groups.”

P171-174

“Post-hoc analysis using Fisher’s exact test with Bonferroni adjustment showed that the prevalence of off-frequency masking (+) ears receiving 101–200 mg/m2 of CDDP was significantly higher than that before CDDP administration (p = 0.015 after Bonferroni adjustment, Cramer's V = 0.31, Fisher’s exact test).”

L196-199

“(p = 0.544 for 125 Hz, p = 0.301 for 500 Hz, p = 1.000 for 1000 Hz, p = 0.652 for 2000 Hz, p = 0.404 for 2000 Hz, p = 0.461 for 3000 Hz, p = 1.000 for 4000 Hz, p = 0.737 for 6000 Hz, and p = 1.000 for 8000 Hz, Fisher’s exact test)”

(This is just a mistake)

Audiometry is a behavioural test. I wonder whether people undergoing chemotherapy may feel generally unwell (they do – I know because I have observed this at close hand over the past 3 months). This may affect their cognitive skills and alertness, etc. in order to complete audiometry. This limitation should be acknowledged. I think a 5 or 10 dB shift is not unreasonable in this situation. On the other hand, I acknowledge that no change was observed in the pure tone audiogram at lower frequencies.

Answer

Please see the answer above.

L229: this sentence needs to be reworded.

Answer

This sentence is reworded.

L268-271

“The other limitation is that inner hair cell damage was not directly proven in ears with positive off-frequency masking. We hypothesized the occurrence of inner hair cell damage after CDDP administration, but injury to other organs in the auditory afferent system can cause similar results.”

L242: It is not clear how this conclusion can be made. It is good to consider the clinical implications, but there was nothing in this study that looked at better management of patients. I suggest it be removed.

Answer

Thank you for the advice. I removed this sentence.

L281

“Overall the study will help in the efficient diagnosis and better management of patients with hearing disorders.”

Reviewer #2: The present paper is a well conducted study covering a very very specific topic.

I would only recommend a review of the English language both for typos and to clarify some lines: lines 181-182; line 213 double "the"; line 224 "fibers is"; 232 precisely

Answer

Thank you for the suggestion. The manuscript was re-submitted to a professional proofreader.

 

Other

Fig 1. Legend

The averaged audiogram before the administration of cisplatin showed a down-sloping type sensorineural hearing loss. The pure-tone thresholds deteriorated significantly with the cumulative dose of cisplatin in the 6000–8000 Hz range.

---

## [Decision Letter · Decision Letter 1]

5 Jun 2023

Systemic cisplatin increases the number of patients showing positive off-frequency masking audiometry

PONE-D-23-02457R1

Dear Dr. Hiraumi,

We’re pleased to inform you that your manuscript has been judged scientifically suitable for publication and will be formally accepted for publication once it meets all outstanding technical requirements.

Kind regards,

Antonino Maniaci

Academic Editor

PLOS ONE

Additional Editor Comments (optional):

Reviewers' comments:

Reviewer's Responses to Questions

**Comments to the Author**

1. If the authors have adequately addressed your comments raised in a previous round of review and you feel that this manuscript is now acceptable for publication, you may indicate that here to bypass the “Comments to the Author” section, enter your conflict of interest statement in the “Confidential to Editor” section, and submit your "Accept" recommendation.

Reviewer #3: (No Response)

2. Is the manuscript technically sound, and do the data support the conclusions?

Reviewer #3: (No Response)

3. Has the statistical analysis been performed appropriately and rigorously? 

Reviewer #3: (No Response)

4. Have the authors made all data underlying the findings in their manuscript fully available?

Reviewer #3: (No Response)

5. Is the manuscript presented in an intelligible fashion and written in standard English?

Reviewer #3: (No Response)

6. Review Comments to the Author

Reviewer #3: I would like to thank the authors for their submission and allowing me to review their work.

This is an interesting study on an important topic. However, I would be grateful if you could add further explanations and changes on the following points:

1) Page 2, line 17

Please specify the age range, gender and mean age (± standard deviation) of the study population.

2) Page 2, line 17

Where was the study conducted?

3) Page 6, line 93

Please add ± standard deviation.

4) Page 6, line 94

Please add more specific demographic and clinical information about the study participants.

5) Page 7, line 109

Where were the audiological tests performed?

6) Page 15, line 253

The “limitations section” should be expanded (e.g. limited number of participants, lack of specific tests such as auditory brainstem responses,…).

7) Page 16, line 273

Which are the future prospects of this study?

8) I would also suggest a review of the English language due to some grammar and phrasing mistakes.

7. PLOS authors have the option to publish the peer review history of their article (what does this mean?). If published, this will include your full peer review and any attached files.

Reviewer #3: No

---

## [Editor Report · Acceptance letter]

26 Jun 2023

PONE-D-23-02457R1 

Systemic cisplatin increases the number of patients showing positive off-frequency masking audiometry 

Dear Dr. Hiraumi:

I'm pleased to inform you that your manuscript has been deemed suitable for publication in PLOS ONE. Congratulations! Your manuscript is now with our production department. 

Kind regards, 

on behalf of

Dr. Antonino Maniaci 

Academic Editor

PLOS ONE